# Development of a Standardized Method for Measuring Bioadhesion and Mucoadhesion That Is Applicable to Various Pharmaceutical Dosage Forms

**DOI:** 10.3390/pharmaceutics14101995

**Published:** 2022-09-21

**Authors:** Lola Amorós-Galicia, Anna Nardi-Ricart, Clara Verdugo-González, Carmen Martina Arroyo-García, Encarna García-Montoya, Pilar Pérez-Lozano, Josep Mª Suñé-Negre, Marc Suñé-Pou

**Affiliations:** 1Department of Pharmacy and Pharmaceutical Technology and Physical Chemistry, Faculty of Pharmacy and Food Sciences, University of Barcelona, Av. Joan XXIII, 27-31, 08028 Barcelona, Spain; 2Pharmacotherapy, Pharmacogenetics and Pharmaceutical Technology Research Group, Bellvitge Biomedical Research Institute (IDIBELL), Av. Gran via de l’Hospitalet, 199-203, 08908 Hospitalet de Llobregat, Spain

**Keywords:** bioadhesion, mucoadhesion, standardized method, pharmaceutical forms, texture analyzer

## Abstract

Although some methods for measuring bioadhesion/mucoadhesion have been proposed, a standardized method is not yet available. This is expected to hinder systematic comparisons of results across studies. This study aimed to design a single/systematic in vitro method for measuring bioadhesion/mucoadhesion that is applicable to various pharmaceutical dosage forms. To this end, we measured the peak force and work of adhesion of minitablets, pellets, and a bioadhesive emulsion using a texture analyzer. Porcine tissue was used to simulate human stomach/skin conditions. The results of these formulations were then compared to those for formulations without the bioadhesive product. We conducted a case study to assess the stability of a bioadhesive emulsion. The results for the two parameters assessed were contact time = 60 s and contact force = 0.5 N at a detachment speed of 0.1 mm/s. Significant differences were observed between the bioadhesive and control formulations, thus demonstrating the adhesive capacity of the bioadhesive formulations. In this way, a systematic method for assessing the bioadhesive capacity of pharmaceutical dosage forms was developed. The method proposed here may enable comparisons of results across studies, i.e., results obtained using the same and different pharmaceutical formulations (in terms of their bioadhesion/mucoadhesion capacity). This method may also facilitate the selection of potentially suitable formulations and adhesive products (in terms of bioadhesive properties).

## 1. Introduction

The term bioadhesion was first introduced in the 1980s when formulations with great retention on biological surfaces started to gain attention. It is defined as the process by which natural and synthetic materials adhere to biological surfaces [1]. Similarly, mucoadhesion (a word derived from bioadhesion) refers to the process by which a bioadhesive substance adheres to the mucosal surfaces of the body [2].

Bioadhesive substances (polymers) are often added to pharmaceutical formulations to enable their adhesion to biological membranes when prolonged contact on the skin is desired [3]. An advantage of increasing the retention time of formulations is that API absorption by biological membranes is enhanced. Thus, pharmacological treatments require less reapplication to be effective, which may increase user compliance. Bioadhesive formulations require a smaller amount of API to ensure a stable therapeutic concentration. This is an advantage, particularly if the therapeutic effect is achieved systemically, since fluctuations in plasma API concentration may cause toxicity. Thus, bioadhesive formulations offer a precise systemic API concentration compared to non-bioadhesive formulations, where values below and above the therapeutic range may occur [4].

Polymers are three-dimensional structures that crosslink and increase in volume in the presence of solvents. Several forces affect the formation of polymer structures that, in turn, enable bioadhesion. The most common are covalent bonds, as well as physical entanglement, ionic forces, hydrophilic interactions, and van der Waals forces [5]. The implications of polymers (as bioadhesive excipients) in formulations are vital in medicine. Bioadhesive formulations are not only beneficial for drug delivery but also for dental (e.g., reattachment of tooth fragments) and surgical treatments (e.g., attachment of a surgical mesh to the peritoneum using fibrin glue), as is noted in the literature [6].

In drug delivery, the most common application sites include dermal, buccal, peroral, nasal, ocular, rectal, and vaginal, and the pharmacologic effect may be local or systemic. The conditions of the various application sites may differ substantially from each other. For example, the gastric mucosa differs from the epidermis. The intestinal epithelium has a mucosal layer (which is mainly composed of water and is in constant contact with an acidic medium, pH 1.2). The skin epidermis, however, has a dry environment and is composed of a lipidic barrier consisting of ceramides, cholesterol, and fatty acids [7]. Since these differences impact the measurement of bioadhesion, the test conditions should be as close as possible to the application site to simulate actual conditions. The animal selected for the bioadhesion test is also an important factor. Pig/rat mucosa and excised vaginal skin (from cow/pig) are generally preferred and are suggested in the literature [4].

Various methods have been used to assess the degree of bioadhesion of finished products or excipients, including both in vitro and in vivo methods. In vitro methods are generally preferred, as they are cost-effective, relatively easy to perform, and less time-consuming. They are often used to screen bioadhesive excipients prior to formulation development or to test potential bioadhesive products with different bioadhesive agents [4].

The vertical detachment strength test is a commonly employed in vitro test. This test can be employed by means of modified balance, a tensile device, a dynamic contact angle analyzer, or an electromagnetic transducer system (though a texture analyzer is perhaps the most employed method) [2,4]. This test quantifies the strength needed to break the internal forces binding the material to the biological surface; that is, to detach the material from the biological surface. Two parameters are commonly measured: detachment or peak force and work of adhesion. Detachment force is the maximal force required to detach the surface from the bioadhesive material. The work of adhesion is calculated from the area of the force–distance curve, following the contact of the bioadhesive material and the biological surface under a constant force during a fixed time. However, some critical factors can influence results and negatively impact the standardization of a method. Parameters used to assess bioadhesive capacity may differ across studies, which makes comparisons between different formulations and adhesive polymers difficult [2]. Some of the critical parameters mentioned in the literature include (1) contact time, (2) the force applied, (3) detachment speed [2], and (4) amount of test material [8] as potential factors affecting bioadhesion.

Some researchers have addressed some of the critical issues related to the development of an optimal standardized method of measuring bioadhesion. Hägeström et al. [8]*,* for instance, concluded that a small amount of bioadhesive gel in contact with two mucosa sheets was preferred to a large volume of bioadhesive gel in contact with one piece of the mucosa. They also concluded, after testing different detachment speeds (0.1–0.5 mm/s), that a lower speed of 0.1 mm/s led to higher precision in terms of detachment force and work of adhesion compared with 0.5 mm/s [8]. However, although some studies [3,9] have suggested that the work of adhesion may have a higher predictive value compared to peak force, the opposite seems to be true for small samples, for which the literature reports that detachment force is more determinant [8]. Furthermore, although some studies [3,8] have been conducted on the bioadhesive capacity of topical forms, the focus has mainly been on intestinal mucoadhesion rather than on the skin [5]. These studies [3,8] have suggested methods for measuring bioadhesion. However, they have predominantly focused on examining the bioadhesive behavior of a single pharmaceutical dosage form. A standardized method to assess bioadhesion is not yet available [8]. In particular, a study with a defined setup and test conditions would enable the development of a systematic method for measuring bioadhesion and mucoadhesion that is applicable to a variety of pharmaceutical dosage forms.

Accordingly, this study sought to design an in vitro method for measuring bioadhesion and mucoadhesion that is applicable to a variety of pharmaceutical dosage forms. In particular, this method aims to simulate the conditions of the human topical and gastric environments.

## 2. Materials and Methods

### 2.1. Materials

The following materials were used: a bioadhesive emulsion, bioadhesive minitablets and pellets, and control minitablets and pellets (with no adhesive properties). Pig ear skin and porcine stomach (obtained from the animal facility of the University of Barcelona, Bellvitge campus) were used as substrates. The following instruments were used to dissect the skin: a disposal sterile scalpel (Sheffield Morton, Sheffield S6 2BJ, England), tweezers (JP Selecta), and scissors. Distilled water was used in the saline solution (0.1% NaCl). An acidic solution with HCl was used to obtain a pH of 1.2. A material texture analyzer (and software) was also used in the experiment: MT-LQ (Stable Micro Systems, Surrey, England).

### 2.2. Formulations

#### 2.2.1. Pellets

Non-bioadhesive pellets were formulated according to Table 1.

To formulate bioadhesive pellets, the non-bioadhesive pellets listed in Table 1 were coated in a fluidized bed with a double coating of 10% polyacrylic acid. See Table 2 for details. 

#### 2.2.2. Minitablets

Non-bioadhesive (no film coating placebo minitablets) and bioadhesive minitablets (film-coated placebo minitablets with bioadhesive polyacrylic acid) were formulated with the indredients in Table 3 and Table 4, respectively.

#### 2.2.3. Emulsions

A non-bioadhesive emulsion and a bioadhesive emulsion were formulated as stipulated in Table 5.

The ingredients in phase A were heated to 75 °C and mixed under stirring with a helix stirrer until complete homogenization. Phase B was heated to 75 °C and mixed under stirring in a separate beaker until homogenization. Phase B was added to phase A under stirring until complete homogenization with turrax and then cooled down to room temperature. Phase C was added slowly under stirring (Table 5). Finally, the pH was adjusted to between 5.5 and 6.5.

### 2.3. Gastric Mucoadhesion

The porcine stomach was defrosted and put into a saline solution (0.1% NaCl) 24 h before the start of the experiment. The stomach mucosa was cut into similar portions and placed on a Petri dish along with the saline solution. Care was taken to ensure that the contact area was completely smooth. Another piece of stomach mucosa was attached to the lower end of the cylindrical probe by means of a piece of cellulose paper and a rubber ring. The cylindrical probe measured 1 cm in diameter and was oriented downward, facing the stomach piece in the Petri dish. Excess saline was withdrawn from the Petri dish, and ten pellets of similar size and weight (87 mg ± 5%) were placed on top of the stomach mucosa. Furthermore, 1 mL of the acid solution was added to the mucosa and pellets. The Petri dish (containing the mucosa and pellets) was placed at the bottom of the texture analyzer.

The same procedure was repeated for the minitablets (ten minitablets of similar size and weight (49.5 mg ± 5%) were used in each experiment).

### 2.4. Topical Bioadhesion

For the topical bioadhesion test, a bioadhesive emulsion (emulsion containing a bioadhesive polymer) was compared with a non-bioadhesive emulsion (emulsion without a bioadhesive polymer).

Pig ear skin (as a substrate simulating human skin in vitro) was used to assess bioadhesion. This choice was informed by reported evidence that it is the most effective in vitro substrate to simulate human skin in terms of its histological and physiological properties [10]. The ears of young pigs (40 kg) were collected from a laboratory animal facility (University of Barcelona, Bellvitge campus). They were cleaned with water at room temperature, and the hairs were trimmed. A scalpel was used to separate the skin (epidermis and dermis) from the pig ear cartilage. The ears were stored in a freezer at −20 °C for two weeks. 

The skins were defrosted 24 h before the start of the study. They were cut into square pieces (3 × 3 cm) and placed on Petri dishes. These skin portions were the substrates for the sample vehicles. A total of 80 mg of the emulsion was spread homogeneously on the substrate skin sheets. In addition, another skin sheet was attached to the lower end of a cylindrical probe (1 cm in diameter) facing downward, opposite the substrate skin with a rubber ring (attached skin) (Figure 1). 

### 2.5. Texture Analyzer

A texture analyzer was used to measure the bioadhesion of pellets, minitablets, and emulsions. This device is commonly used to measure bioadhesion/mucoadhesion [11]. The method and experiment settings are described in Section 2.5.1 (and the preliminary method settings are described in Section 2.5.2).

#### 2.5.1. Experimental Settings

The upper part of the texture analyzer (with the attached skin) was placed as close as possible to the substrate skin. Contact was avoided between the two skin sheets (porcine stomach for the pellets and minitablets and porcine ear skin for the emulsions). In this position, the texture analyzer was lowered to 0.1 mm/s until contact between the substrate skin and the attached skin was made. The triggering force (by which the contact with the sample was calculated) was 0.01 N. The two skin pieces were in contact for 60 s under a force of 0.5 N. The upper part of the texture analyzer was lifted at a speed of 0.1 mm/s until the separation of the two skin sheets occurred. Figure 1 illustrates the peak force and the work of adhesion, which are the force needed to separate the two skin sheets and the area under the force–distance curve, respectively.

#### 2.5.2. Method Development 

Preliminary experiments were conducted to establish the best conditions for performing the test described in Section 2.5.1. Porcine stomach was used as a substrate. Three critical parameters were selected to determine the conditions of the test: (1) the contact time between the attached and the substrate skin, (2) the force exerted during contact time, and (3) the detachment speed. The test conditions for contact time were 15, 20, 60, and 900 s. The test conditions for contact force were 0.3, 0.5, and 1 N. The detachment speed was set to 0.1 mm/s according to Hägerström et al. [3]. Hägerström et al. studied different speeds and concluded that the one that gave the best discriminative values was 0.1 mm/s. The findings of this author on the detachment speed were taken into account in the present article.

The test was performed at controled room temperature (25 ± 2 °C) according to the Eur Ph [12].

The peak force of ten pellets with the bioadhesion film was compared with the same number of pellets without the bioadhesion film. The increase difference was determined by the percentage increase of the two types of pellets and calculated with Equation (1).
(1)$$Percentage\;increase\;\left(\%\right)=\frac{{\bar{F}}_{\left(Film-No\;film\right)}}{{\bar{F}}_{No\;film}}\times 100$$
where *F* stands for the peak force (N) of the average results obtained for the pellets with the bioadhesive film and those without the bioadhesive film. 

The same calculation was used to assess the percentage increase in the work of adhesion (calculated with Equation (2)).
(2)$$Percentage\;increase\;\left(\%\right)=\frac{{\bar{W}}_{\left(Film-No\;film\right)}}{{\bar{W}}_{No\;film}}\times 100$$
where *W* stands for the work of adhesion (mJ) of the average results obtained for the pellets with the bioadhesive film and those without the bioadhesive film.

After establishing the method for the pellets, Equations (1) and (2) were used to calculate the force and work percentage increase of the minitablets and emulsions.

A stability test was also performed for the bioadhesive emulsion. The bioadhesive emulsion was assessed immediately after and one year after manufacturing. The bioadhesive behavior of the bioadhesive emulsions (1 and 2) was then compared with (1) a non-bioadhesive emulsion and (2) the substrate skin alone. Moreover, the bioadhesive emulsion was tested in a wet environment so that the bioadhesive behavior could be assessed following exposure to water. For this purpose, the skin was moistened with 5 mL of distilled water (following the application of the bioadhesive emulsion to the skin).

### 2.6. Statistical Analysis

The bioadhesive emulsions were assessed five times (*n* = 5), and the following emulsions were assessed three times (*n* = 3): non-bioadhesive emulsions, bioadhesive emulsions after 1 year, and bioadhesive emulsions at wet conditions. Bioadhesive and control pellets were assessed nine times (*n* = 9), and bioadhesive minitablets and control minitablets were assessed 6 times (*n* = 6). Statistical relevance was established at *p* < 0.05 for all statistical tests performed. GraphPad Prism version 9.0.0 for Windows Harvey Motulsky (GraphPad Software, San Diego, CA, USA) was used as a statistical tool. Data are expressed as mean ± SD, and statistical significance was measured using the unpaired *t*-test (two measurement datasets), one-way ANOVA, and multiple comparison tests (more than two datasets). 

## 3. Results 

### 3.1. Parameters Analysis

Unlike trigger force, withdrawal speed, contact force, and contact time have been reported as key factors influencing the design of an in vitro method for measuring bioadhesion and mucoadhesion [13]. To determine the best conditions for measuring bioadhesion using a texture analyzer, the current study combined some of the factors that could affect it. In particular, the velocity was set to 0.1 mm/s as a result of findings in the literature [3] that 0.1 mm/s led to less variability in the measurements.

As has previously been described, the bioadhesion of ten pellets (ten with and ten without the bioadhesive film) was compared. Three different contact forces were applied: 0.3, 0.5, and 1 N. While the bioadhesive pellets showed higher force peaks at 0.3 and 0.5 N contact forces, at 1 N, the opposite effect was observed, i.e., a negative percentage increase. Of the two other forces tested, 0.5 N showed a higher force value. Although the percentage differences were small, the bioadhesive pellets at 0.5 N showed a marginally higher force value compared with the non-bioadhesive pellets.

Having determined the contact force as 0.5 N, the contact time was assessed. A force of 0.5 N was applied at 0.01 mm/s in this experiment. The following contact times were assessed: two short contact times (15 s and 20 s), an intermediate contact time of 60 s, and a long contact time of 900 s. Among these, 15 s was immediately discarded, as the percentage increase was negative. However, a positive percentage increase was obtained with the 20 s test, although the results showed large deviations. Additionally, a high deviation of the results was observed with the high contact time (900 s). The contact time of 60 s was thus selected as the most suitable contact time for testing, as this contact time showed positive percentage increase values with only small deviations (Figure 2).

### 3.2. Gastric Mucoadhesion and Statistical Study

#### 3.2.1. Pellets

The results in Table 6 show that there are significant differences in terms of both studied parameters (peak and work forces) between the bioadhesive and non-bioadhesive pellets. The difference between both types of pellets was 0.048 N in terms of peak force and 0.119 mJ in terms of work force. The similarity in the peak force and work force of both types of pellets was also reflected in the percentage increase (187.8% and 179.8%, respectively). Thus, both parameters confirmed bioadhesion (Figure 3, with raw data being listed in Table A1).

It is worth mentioning that the relatively high standard deviation (±0.053) in the force of the bioadhesive pellets was the result of a single value exceeding the mean (Figure 3a,c). However, this was not observed in the work of adhesion (Figure 3b,d). Moreover, the results of the work of adhesion were more dispersed for both pellet types compared with the peak force values.

#### 3.2.2. Minitablets

Greater differences were observed for the work of adhesion compared with the peak force between the two types of minitablets (with film and no film) (Table 7). While the difference in the peak force for both minitablet types was 0.015 N, the difference in the work force was 0.065 mJ. This represented more than twice the increase in the work of adhesion compared with the peak force of adhesion (60% peak force increase compared with 141.3% work increase). The raw data are listed in Table A2.

Again, the work of adhesion showed a higher statistical value compared with the peak force. However, both parameters were optimal predictors for assessing the bioadhesion of the minitablets. Therefore, despite the differences in the magnitude of the increase, both parameters showed high statistical differences for the two minitablet types with *p* < 0.01 and *p* < 0.001 for peak force and work of adhesion, respectively (Figure 4a,b). This difference in bioadhesive results for both parameters (peak force and work of adhesion) is shown in the box plots; peak force and work of adhesion results for the second and third quartiles are separated from each other (Figure 4b,d).

### 3.3. Emulsions

The peak force and work of adhesion results for both the bioadhesive (BE) and the non-bioadhesive (non-BE) emulsions are shown in Figure 5a–d and Table 3. While the average peak force difference between both emulsion types was 0.283 N, the average work of adhesion difference between both emulsions was 0.375 mJ. The peak force of the bioadhesive emulsion resulted in more than three times the peak force of the non-bioadhesive emulsion. However, the bioadhesive emulsion increased more than 20 times compared with the non-BE for the work of adhesion.

The percentage increase in the work of adhesion was nearly eight times higher compared with the percentage increase in the peak force (241.9% peak force vs. 2205% work force, Table 8). The raw data are listed in Table A3.

Despite these differences in the magnitudes of the increase, both parameters were equally valid for predicting the bioadhesion of emulsions. This was because the peak force and work of adhesion of the bioadhesive emulsions were significantly different from the non-bioadhesive ones with *p*-values of < 0.001 and 0.01, respectively (Figure 5a,b). The high statistical significance is visible in Figure 5c,d; the box plots of the bioadhesive emulsions do not overlap with those of the non-bioadhesive ones for either the peak force or work of adhesion.

Regarding the precision of the measurements, the peak force had a smaller standard deviation for the bioadhesive emulsion compared with the work of adhesion. However, this effect was the opposite for the non-bioadhesive emulsion, where the standard deviation for the work of adhesion was smaller.

### 3.4. Case Study

The results of the bioadhesive test are presented in Table 9 and Figure 6a–d. The peak force and work of adhesion were measured for each emulsion type. The results showed that there were significant differences (*p* ≤ 0.05) between the bioadhesive emulsion and the mock (skin without emulsion) in terms of both the peak force and work of adhesion. As was mentioned in Section 3.2, the bioadhesive emulsion was statistically different to the non-bioadhesive emulsion. Additionally, no significant difference was observed between the mock and the non-bioadhesive emulsion (Table 4). The raw data are listed in Table A3. 

As has previously been stated, the bioadhesive emulsion was then compared with the same emulsion after one year of storage. Figure 6a–d show very similar peak forces and work of adhesion after one year of storage. The peak force of the bioadhesive emulsion was originally 0.40 ± 0.056 N, and after one year the peak force was 0.377 ± 0.045 N. There was only a 0.023 N difference after one year, which represents roughly a 5% decrease. Additionally, the bioadhesive emulsion that had been manufactured one year earlier (BE 1 year) was compared with the mock and the non-bioadhesive emulsion (non-BE). Both the mock and the non-bioadhesive emulsion showed statistical differences compared with BE 1 year. While the differences observed for the mock in terms of peak force and work of adhesion were 0.297 N and 0.308 mJ (a 371% and 2803% increase, respectively), the difference observed for the non-bioadhesive emulsion was 0.260 N and 0.302 mJ (a 222.3% and 1776% increase, respectively). We can observe that the percentage increases were similar for peak force and greater for work of adhesion. In addition, the results of the percentage increase obtained for work of adhesion showed higher standard deviations than those obtained for peak force.

This measurement precision is shown in the box plots (Figure 6c,d). The results on the work of adhesion were more spread out, especially for the emulsions showing bioadhesive behavior: BE and BE 1 year (* in Figure 6a,b). The results for bioadhesive emulsions did not overlap with the results for the non-bioadhesive emulsion or the mock.

Finally, the moistened bioadhesive emulsion was compared with the non-bioadhesive emulsion and mock in terms of peak force and work of adhesion. There was no significant difference between the bioadhesive emulsion that was moistened with 5 mL of water and the mock in relation to peak force and work of adhesion, as is shown in Figure 6a,b. This confirms that the bioadhesive emulsion that was moistened with 5 mL was not bioadhesive. The percentage increase was higher for the work of adhesion compared with the peak force: 638.8% and 165% compared with the mock and 376% and 81.2%, respectively, compared with the non-BE.

## 4. Discussions

As was mentioned in the introduction (Section 1), although some methods for measuring bioadhesion/mucoadhesion have been proposed, a standardized method has not been identified in the literature. This is expected to hinder systematic comparisons of results across studies.

In particular, most of the published studies on bioadhesion have been performed using mucosa. The choice of substrate usually depends on the route of administration of the product. In cases where the product is intended for oral, nasal, or intravaginal use, the use of mucosal tissue is the norm, and numerous studies have described the development of these bioadhesive products [4]. However, few studies have addressed bioadhesion for skin administration. This external part of the body may be a target for semisolid formulations (with bioadhesive properties carrying one or more active substances). The methods for bioadhesion have been described in the literature for both the mucosa and the skin, but separately. However, a method compatible with different pharmaceutical dosage forms and skin/mucosa substrates has not yet been established. Therefore, as previously stated, this study proceeded to select two solid products for oral administration and a semisolid form for topical administration in an attempt to develop an in vitro method for measuring bioadhesion and mucoadhesion that is applicable to a variety of pharmaceutical dosage forms.

The bioadhesive product was compared with a non-bioadhesive formulation, in contrast to other studies [3,8,14] in which the substrate without any product was the mock. Thus, the bioadhesive material itself was assessed as the formulation without the bioadhesive film may have an independent measure of adhesion. However, as was demonstrated in the case study (Section 3.3), there were only small differences in peak force and work of adhesion between the skin without emulsion and the non-bioadhesive formulation. Therefore, both skin without emulsion and non-bioadhesive emulsion were deemed valid mocks.

Regarding the product use, it was important to assess bioadhesion with amounts as close as possible to the actual conditions of use. While package leaflets do not establish single-dose prescriptions for topical formulations, a patent for a bioadhesive gel product containing acyclovir selected 8.3 mg/cm^2^ as the appropriate amount of gel product for topical use [15]. However, sunscreen products have clear standards for measuring sun protection factors (SPF) in vivo and do specify an amount of 2 mg/cm^2^ to achieve the labeled SPF [16]. This is equivalent to 50 mg of the product homogeneously spread on a 5 × 5 cm skin sheet section. In the present study, 80 mg of the emulsion was spread on 3 × 3 cm skin sheet sections. This corresponds to 5.3 mg/cm^2^. This amount was selected because it lies between 2 mg/cm^2^ and 8.3 mg/cm^2^. The amount used in this study is therefore closer to that of actual applications of topical formulations compared with other studies in the same field [3,8,17,18].

The parameters of peak force and work of adhesion were both valid for determining the bioadhesion of the formulation. Significant differences were observed between bioadhesive and non-bioadhesive formulations when any of these parameters were used. The high standard deviation of the work of adhesion can be explained by the fact that the work of adhesion is the result of two factors, namely force and distance, whereas peak force is a direct measure. Furthermore, contact time and contact force were determining factors for bioadhesion. 

The method proposed here was found to work on formulations of different natures, namely, solid formulations (minitablets and pellets) and semi-solid formulations (emulsions). Minitablets and pellets were chosen because they are representatives of solid formulations, and an emulsion is representative of a semi-solid formulation. It should, however, be emphasized that the test must be performed under the same conditions for all measurements as minor changes may result in variations. For instance, in our study, the bioadhesive emulsion was moistened with 5 mL of water prior to the bioadhesive test, and consequently, the bioadhesion decreased compared with the unmoistened bioadhesive emulsion. This further illustrates the challenge of extrapolating results performed under different settings.

## 5. Conclusions

A systematic method for measuring the bioadhesive capacity of pharmaceutical dosage forms (for topical and internal mucosa applications) was successfully developed. The method proposed here may enable the comparison of results across studies, i.e., results obtained using the same and different pharmaceutical formulations (in terms of bioadhesion/mucoadhesion capacity). This method could also facilitate the selection of potentially suitable formulations and adhesive products (in terms of their bioadhesive properties).

Future research may wish to further verify the applicability of such a method to other pharmaceutical dosage forms (e.g., gels, lipogels, emulgels, patches, and capsules).

## Figures and Tables

**Figure 1 pharmaceutics-14-01995-f001:**
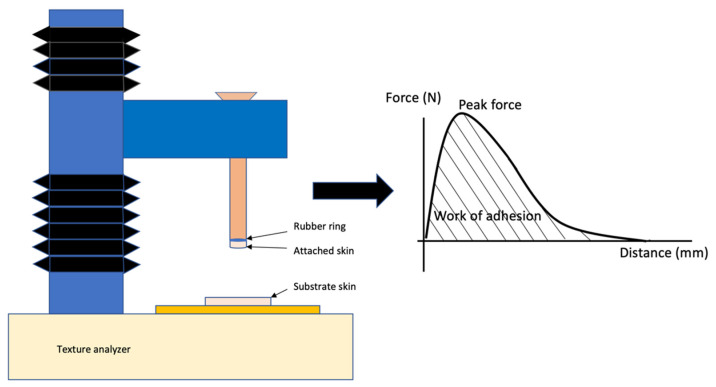
Experimental settings: Schematic illustration of the texture analyzer. The peak force and work of adhesion provided by the texture analyzer software Materials Master (SET19002, Stable Micro Systems, Surrey, England). Illustration based on Carvalho et al., 2013 [5] and Hägeström et al., 2004 [8].

**Figure 2 pharmaceutics-14-01995-f002:**
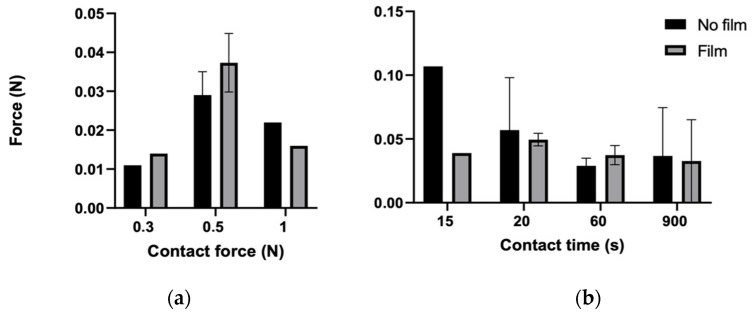
Adhesion force applied to pellets with (**a**) different contact forces and (**b**) different contact times. (**c**) Percentage increases at different contact forces and (**d**) percentage increases at different contact times.

**Figure 3 pharmaceutics-14-01995-f003:**
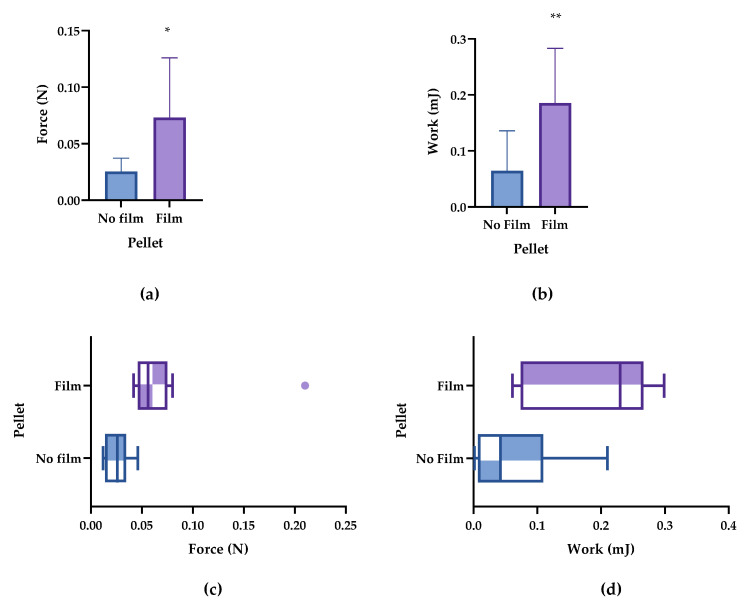
Bioadhesion parameters of the bioadhesive and non-bioadhesive pellets. (**a**) Peak adhesion force and (**b**) work of adhesion. (**c**) Box plot of the peak of adhesion and (**d**) work of adhesion. * *p* < 0.5, ** *p* < 0.01.

**Figure 4 pharmaceutics-14-01995-f004:**
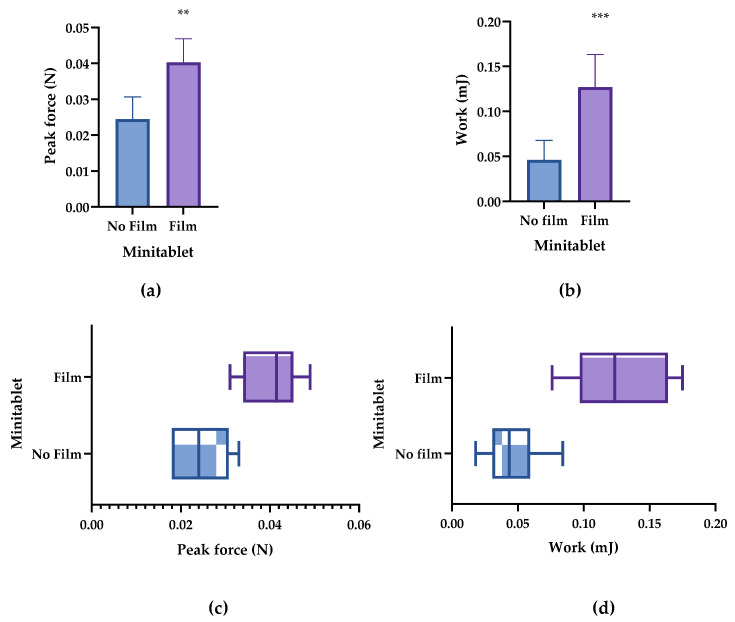
Bioadhesion parameters of the bioadhesive and non-bioadhesive minitablets. (**a**) Peak force and (**b**) work of adhesion. (**c**) Box plot of the peak force and (**d**) work of adhesion. ** *p* < 0.01, and *** *p* < 0.001.

**Figure 5 pharmaceutics-14-01995-f005:**
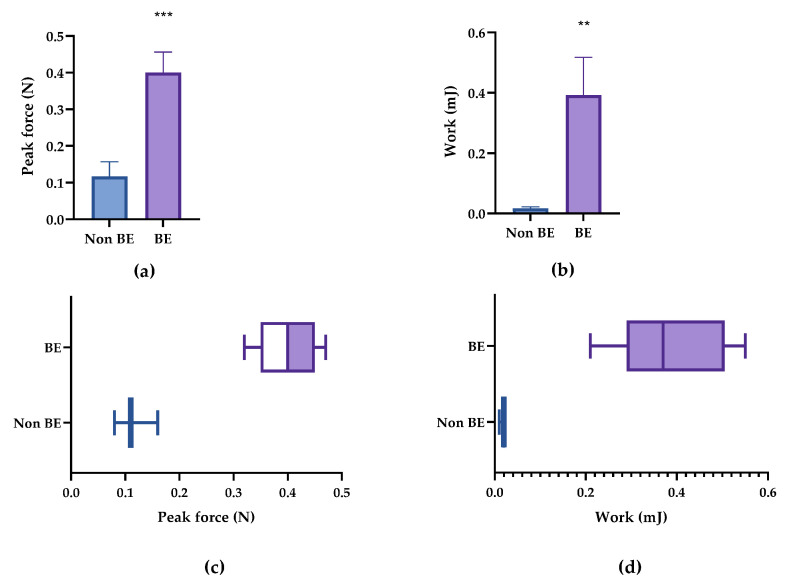
Bioadhesion parameters of the bioadhesive and non-bioadhesive emulsions (BE and non-BE). (**a**) Peak and (**b**) work of adhesion. (**c**) Box plot of the peak force and (**d**) work of adhesion. ** *p* < 0.01, and *** *p* < 0.001.

**Figure 6 pharmaceutics-14-01995-f006:**
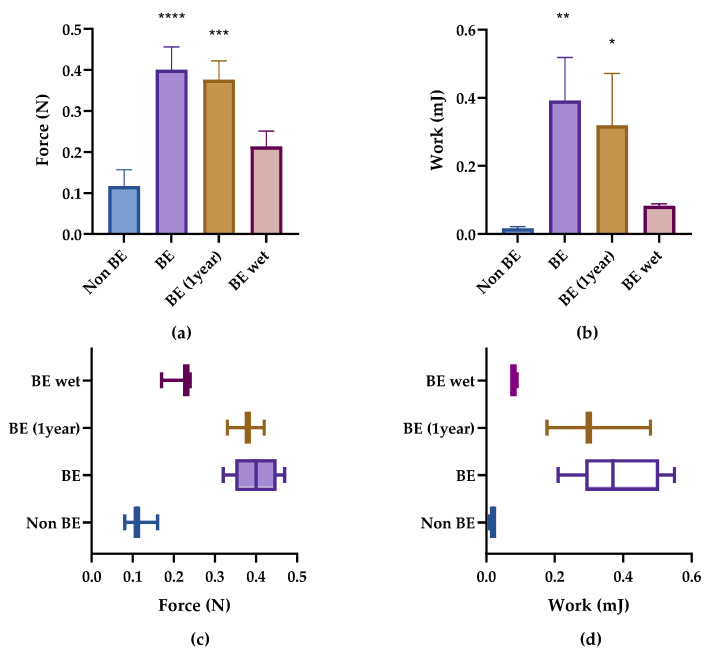
The bioadhesion parameters of the non-bioadhesive emulsion (non-BE), bioadhesive emulsion (BE), bioadhesive emulsion after one year, and bioadhesive emulsion in wet conditions (BE 1 year). (**a**) Peak adhesion force and (**b**) work of adhesion. (**c**) Box plot of the peak of adhesion and (**d**) work of adhesion. * *p* < 0.5, ** *p* < 0.01, and *** *p* < 0.001, **** *p* < 0.0001.

**Table 1 pharmaceutics-14-01995-t001:** Preparation of non-bioadhesive pellets.

Ingredients	%
Vinylpyrrolidone–vinyl acetate copolymer	5.0
Hypromellose	5.0
Sodium bibarbonate	30.0
Barium sulfate	20.0
Microcrystalline cellulose	40.0

**Table 2 pharmaceutics-14-01995-t002:** Preparation of bioadhesive pellets.

Ingredients	%
Pellets in Table 1	70
Acrylic acid polymer solution (10%) (polycarbophil USP)	30

**Table 3 pharmaceutics-14-01995-t003:** Preparation of non-bioadhesive minitablets.

Ingredients	%
Sodium croscarmellose	2.0
Magnesium stearate	2.0
Talc	4.0
Isomaltose	ad 100

**Table 4 pharmaceutics-14-01995-t004:** Preparation of bioadhesive minitablets.

Ingredients	%
Non-bioadhesive minitablets	98.0
Acrylic acid polymer solution (1%) (polycarbophil USP)	2.0

**Table 5 pharmaceutics-14-01995-t005:** Preparation of bioadhesive emulsions.

Phase	Ingredients	%
A	Behenyl alcohol	3.0
	Caprylic/capric trigliceride	3.0
	Dodecyl benzoate	3.0
	7-methyloctanoate	4.0
	Phenoxyethanol	0.8
	Benzoic acid 2-ethylhexyl ester	5.0
B	Distilled water	ad 100
	Disodium EDTA	1.0
	Tris(hydroxymethyl)aminomethane	2.4
	Glycerine	0.8
	Acrylic acid polymer (polycarbophil USP)	X ^1^
C	Ethylhexyloxy hydroxyphenyl methoxyphenyl triazine	7.0
	Benzotriazolyl tetramethylbutylphenol	7.0
	Ethanol	10.0
	Propylenglycol	24.0

^1^ x = 0, 1.

**Table 6 pharmaceutics-14-01995-t006:** Peak force (F) and work of adhesion (W) of pellets without the mucoadhesive film (control) and with the mucoadhesive film. Differences between the two groups shown as mean ± SD, (*n* = 9).

Pellets	F (N)	W (mJ)
No film	0.025 ± 0.012	0.066 ± 0.070
Film	0.073 ± 0.053	0.185 ± 0.135
Difference (film−no film)	0.048	0.119
Increase (%)	187.8	179.8

**Table 7 pharmaceutics-14-01995-t007:** Peak force (F) and work of adhesion (W) of minitablets without the mucoadhesive film (control) and with the mucoadhesive film. Differences between the two groups shown as mean ± SD, (*n* = 6).

Minitablets	F (N)	W (mJ)
No film	0.025 ± 0.006	0.046 ± 0.022
Film	0.040 ± 0.007	0.111 ± 0.060
Difference (film−no film)	0.015	0.065
Increase (%)	60	141.3

**Table 8 pharmaceutics-14-01995-t008:** Peak force (F) and work of adhesion (W) of the bioadhesive (BE) and non-bioadhesive (Non-BE) emulsions. Differences between the two groups shown as mean ± SD, (*n* = 5 for BE and *n* = 3 for non-BE).

Samples	F (N)	W (mJ)
Non-BE	0.117 ± 0.040	0.017 ± 0.009
BE	0.400 ± 0.056	0.392 ± 0.126
Difference (BE−Non-BE)	0.283	0.375
Increase (%)	241.9	2205.9

**Table 9 pharmaceutics-14-01995-t009:** Peak force (F) and work of adhesion (W) of the bioadhesive (BE), non-bioadhesive (Non-BE) emulsions, BE after one year (ba.em-1y) and Bioadhesive emulsion wet (ba.em-wet). Differences between the groups shown as mean ± SD, (*n* = 5 for BE and *n* = 3 for non-BE, ba.em-1y and ba.em-wet).

Samples	F (N)	W (mJ)
Mock	0.080	0.011
Non-BE	0.117 ± 0.040	0.017 ± 0.009
BE	0.400 ± 0.056	0.392 ± 0.126
Difference (BE−mock)	0.320	0.381
Increase (%)	400.0	3463.6
Difference (BE−Non-BE)	0.283	0.375
Increase (%)	241.8	2205.9
BE after 1 year	0.377 ± 0.045	0.319 ± 0.152
Difference (ba.em-1y−mock)	0.297	0.308
Increase (%)	371.3	2803.0
Difference (ba.em-1y−Non-BE)	0.260	0.302
Increase (%)	222.2	1776.0
Bioadhesive emulsion wet	0.212 ± 0.041	0.081 ± 0.004
Difference (ba. em-wet−mock)	0.132	0.070
Increase (%)	165.0	638.8
Difference (ba. em-wet−Non-BE)	0.095	0.064
Increase (%)	81.2	376.0

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
