# Peer review of "Development of a Standardized Method for Measuring Bioadhesion and Mucoadhesion That Is Applicable to Various Pharmaceutical Dosage Forms"

_pharmaceutics, 2022, doi:10.3390/pharmaceutics14101995_

Round 1
Reviewer 1 Report
The author chooses three parameters (contact time, contact force, temperature) to develop a systematic method for bioadhesive capacity of the pharmaceutical forms for different porcine tissues (porcine stomach and skin), while detachment speed is also an important parameter. Most biological tissues are viscoelastic, so different separation speed would change measures values.
The authors also claim temperature is an important parameter but did not show the summary figure/data in the text, for the ex vivo testing, temperature may be less important than the detachment speed. Suggest that remove the temperature factor and consider detachment speed as another key parameter to set up the standardized method.
Reviewer 2 Report
The manuscript entitled “Development of a standardized method for measuring bioadhesion and mucoadhesion, applicable to various pharmaceutical forms” reports the tests the authors performed in the attempt to standardize a method to measure bio/muco-adhesion of different pharmaceutical dosage forms in vitro.
Although the topic is worth investigation, the manuscript does not report sufficient data to support the reliability of the developed method. The authors tested only three formulations without clearly describing the experimental protocol they used (e.g. the amount of each formulation used to perform the tests is not indicated, the composition of tested formulations is not reported, the content and type of bio-adhesive polymer in the formulation is not specified, etc.). Many issues should be addressed. Why did the authors choose the formulations tested? Would the proposed method provide reliable results using different bio-adhesive polymers or different concentrations of the same bio-adhesive polymer?
Round 2
Reviewer 2 Report
The authors did not revise the manuscript properly. The authors reported the composition of the tested emulsion in Table 5. The heading of this Table (Preparation of bioadhesive minitablets) is incorrect. The authors reported the emulsion composition without specifying the type of non-ionic emulsifier, wax-type emulsifier, emollient ester, medium chain glycerides, acrylic acid polymer, and alcohol type solvent A and B. The knowledge of the raw materials used to prepare such emulsions is fundamental to ensure experiment reproducibility. In addition, the authors added triazone UV filter 5.0 % in the lipid phase and they reported twice triazone UV filter 7.0 % in the water phase. How much triazone UV filter did the authors use and in which phase did they add this substance? Furthermore, the authors did not report the preparation method they used to obtain these emulsions. Such information should be reported in the manuscript.
Round 3
Reviewer 2 Report
The authors revised the manuscript properly.